# Rapid resistance development to three antistaphylococcal therapies in antibiotic-tolerant *staphylococcus aureus* bacteremia

**Christopher R. Miller[1], Jonathan M. Monk[2], Richard Szubin[2], Andrew D. Berti[1,3]** *

**1** Department of Pharmacy Practice, Wayne State University College of Pharmacy and Health Sciences, Detroit, MI, United States of America, **2** Department of Bioengineering, University of California at San Diego, La Jolla, CA, United States of America, **3** Department of Biochemistry, Microbiology and Immunology, Wayne State University College of Medicine, Detroit, MI, United States of America

* andrew.berti@wayne.edu

**Data Availability Statement:** The data used in this study are available from GenBank under the accession number: PRJNA745996 (https://www.ncbi.nlm.nih.gov/bioproject/PRJNA745996/).

## Abstract

Understating how antibiotic tolerance impacts subsequent resistance development in the clinical setting is important to identifying effective therapeutic interventions and prevention measures. This study describes a patient case of methicillin-resistant *Staphylococcus aureus* (MRSA) bacteremia which rapidly developed resistance to three primary MRSA therapies and identifies genetic and metabolic changes selected *in vivo* that are associated with rapid resistance evolution. Index blood cultures displayed susceptibility to all (non-beta-lactam) antibiotics with the exception of trimethoprim/ sulfamethoxazole. One month after initial presentation, during the same encounter, blood cultures were again positive for MRSA, now displaying intermediate resistance to vancomycin and ceftaroline and resistance to daptomycin. Two weeks later, blood cultures were positive for a third time, still intermediate resistant to vancomycin and ceftaroline and resistant to daptomycin. Mutations in *mprF* and *vraT* were common to all multidrug resistant isolates whereas mutations in *tagH*, *agrB* and *saeR* and secondary *mprF* mutation emerged sequentially and transiently resulting in distinct *in vitro* phenotypes. The baseline mutation rate of the patient isolates was unremarkable ruling out the hypermutator phenotype as a contributor to the rapid emergence of resistance. However, the index isolate demonstrated pronounced tolerance to the antibiotic daptomycin, a phenotype that facilitates the subsequent development of resistance during antibiotic exposure. This study exemplifies the capacity of antibiotic-tolerant pathogens to rapidly develop both stable and transient genetic and phenotypic changes, over the course of a single patient encounter.

## Introduction

The emergence of antimicrobial resistance is a well-recognized threat to public health. Decades of research on antibiotic resistance have provided insight into resistance development and reinforced the need for proper antimicrobial stewardship. However, the contribution of antibiotic tolerance is less clear in the clinical setting. Given a sufficiently large bacterial population

**Funding:** This work was supported by departmental funds to ADB from Wayne State University College of Pharmacy and Health Sciences, Department of Pharmacy Practice. CRM is supported by the Research Scholars program at Wayne State University College of Pharmacy and Health Sciences. JMM and RS were funded through the NIH NIAID grant (1-U01-AI124316-01). The funders had no role in study design, data collection and analysis, decision to publish, or preparation of the manuscript.

**Competing interests:** The authors have declared that no competing interests exist.

*in vitro*, some cells enter a distinct, non-dividing metabolic state and are able to survive transient exposure to antibiotics without a corresponding change in the population's minimum inhibitory concentration (MIC) [1]. These tolerant individuals remain genetically identical to the overall population and can replenish a dividing population once antibiotic pressure is removed. Furthermore, a population with a large proportion of antibiotic tolerant bacteria has a proclivity to rapidly develop resistance to antimicrobials *in vitro* [2] and may be associated with persistent infections *in vivo* [3]. While there exist clear parallels between the two phenomena, antibiotic tolerance is distinct from the concept of heteroresistance where a small subpopulation of bacteria exhibits a different resistance profile and is able to continue growth in the presence of antibiotics [4].

The idea that tolerance or heterotolerance facilitates resistance development was proposed as early as the 1980s [5]. This theory has subsequently been validated *in vitro* in a diverse assortment of microorganisms and a diverse variety of antibiotics [6–8]. In addition to maintaining a viable cell reservoir in which mutations can develop, some tolerant microorganisms demonstrate a higher mutation rate which can further drive resistance development [9]. Indeed, a large array of genetic changes can result in an increased prevalence of tolerant microbes within a population, typically by prolonging the "lag phase" of bacterial growth or reducing the exponential growth rate [1]. While this phenomenon has been modeled extensively in both *in vitro* experiments and mathematical modeling, only one report to date clearly describes tolerance contributing to resistance development in patients [10].

In this study, we first present a patient case of methicillin-resistant *Staphylococcus aureus* (MRSA) bacteremia in which antibiotic tolerance facilitated the development of resistance to three anti-staphylococcal therapies over a six-week clinical course. We then analyze the genetic and metabolic evolution of an antibiotic-tolerant isolate of MRSA as it acquired multi-drug resistance *in vivo*.

## Patient case

A male patient in his late-60s presented to our hospital in October 2018 with altered mental status and methicillin-resistant *Staphylococcus aureus* (MRSA) bacteremia. The patient history was significant for end-stage renal disease requiring dialysis, peripheral vascular disease and insulin-dependent diabetes mellitus. Nasal screening for staphylococcal colonization was not performed. The patient was treated for diabetic ketoacidosis in the emergency department and intubated following emergent acute respiratory failure. His intrajugular dialysis catheter was removed. Arteriovenous HeRO grafts were unremarkable on physical exam and negative for fluid accumulation by ultrasound and thus retained. Transesophageal echocardiogram was unremarkable for infective endocarditis. Tagged white blood cell scans failed to identify any foci of infection. He was determined to be a poor surgical candidate for graft revision and managed medically. Initial peripheral blood culture bacteria were MRSA susceptible to ceftaroline (CPT), daptomycin (DAP), linezolid (LZD) and vancomycin (VAN). The patient was treated empirically with cefepime and VAN (hospitalization days 1–3), and subsequently narrowed to CPT and VAN (days 4–13). Blood cultures cleared day 8 and remained clear on subsequent cultures (days 10 and 11). Week two, following concerns for ventilator-associated pneumonia, therapy was escalated to broad-spectrum β-lactam (cefepime) plus DAP (days 14–23) followed by DAP monotherapy (days 23–32). The following month, breakthrough positive cultures were noted on therapy and displayed non-susceptibility (referred to as resistant throughout for ease of presentation) or borderline-resistance to CPT, DAP and VAN. Blood cultures cleared by day 33 and the regimen was switched to CPT plus LZD (days 33–36) and transitioned to CPT plus DAP for the remainder of the encounter (days 36–46). Peripheral

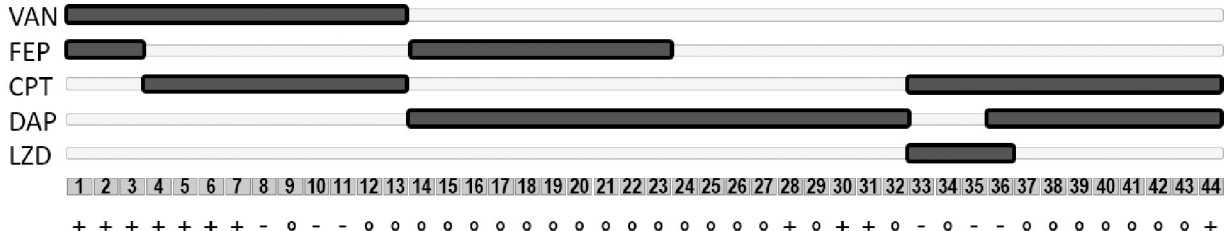

| Collection Date: | Hospital Day 1 | Hospital Day 3 | Hospital Day 7 | Hospital Day 28 | Hospital Day 31 | Hospital Day 44 |
|---|---|---|---|---|---|---|
| Specimen Name: | BSN14S1 | BSN14S2 | — | BSN14R1 | BSN14R2 | BSN14RB |
| Ceftaroline[†] | 0.5 | — | — | 2 | — | — |
| Clindamycin | ≤ 0.5 | ≤ 0.5 | ≤ 0.5 | ≤ 0.5 | ≤ 0.5 | ≤ 0.5 |
| Daptomycin[†] | 0.5 | 0.5 | 0.5 | 4 | 4 | 4 |
| Erythromycin | ≤ 0.5 | ≤ 0.5 | ≤ 0.5 | ≤ 0.5 | ≤ 0.5 | ≤ 0.5 |
| Gentamicin | ≤ 2 | ≤ 2 | ≤ 2 | 2 | 2 | 2 |
| Linezolid | 2 | 2 | 2 | 2 | 2 | ≤ 1 |
| Oxacillin | > 2 | > 2 | > 2 | > 2 | > 2 | > 2 |
| Rifampin | ≤ 0.5 | ≤ 0.5 | ≤ 0.5 | ≤ 0.5 | ≤ 0.5 | ≤ 0.5 |
| Telavancin[†] | — | — | — | 0.12 | — | — |
| Trimethoprim/ Sulfamethoxazole | > 2/38 | > 2/38 | > 2/38 | > 2/38 | > 2/38 | > 2/38 |
| Vancomycin | 1 | 1 | 1 | 2 | 2 | 2 |

**Fig 1. Clinical timeline.** Patient presented to Emergency Department October 2018 and was discharged on hospitalization day 44 to a skilled nursing facility. Symbols indicate days of documented positive blood culture (+), negative blood culture (-) or blood culture not collected (○). Results of susceptibility testing as reported in the patient's electronic medical record are reproduced below. VAN, vancomycin; FEP, cefepime; CPT, ceftaroline; DAP, daptomycin; LZD, linezolid. [†]Values determined by E-test.

blood cultures remained clear until a recurrence day 44. Patient was determined clinically stable for transfer day 46, discharged with positive blood cultures to a skilled nursing facility and lost to follow up. A timeline and characterization of isolates from this patient encounter is provided as Fig 1.

## Results and discussion

### Genomic assessment

All isolates from patient BSN14 were confirmed to be isogenic (2,956,388 bp chromosome, Pulse Field type USA300, Multilocus Sequence Type 8 and spa type t064) by whole genome sequencing, ruling out coinfection or superinfection as potential etiologies. Genetic variations between serial isolates are reported in Table 1. No sequence variations were noted between BSN14S1 and BSN14S2. Relative to BSN14S1, BSN14R1 contained sequence variations in *mprF* (ntC941T, P314L), *vraT* (ntG451A, A151T) and *tagH* (ntG115A, A39T). BSN14R2

**Table 1. Mutational differences between serial isolates identified by whole-genome sequencing.**

| Position | Mutation | Gene | Function | BSN14S1 | BSN14S2 | BSN14R1 | BSN14R2 | BSN14RB |
|---|---|---|---|---|---|---|---|---|
| 710,607 | A39T | *tagH* | Teichoic acid export | | | ✓ | | |
| 780,975..89 | delD46..L50 | *saeR* | Response regulator | | | | | ✓ |
| 1,439,232 | G249Gfs14 | *mprF* | Phosphatidylglycerol lysyltransferase | | | | ✓ | |
| 1,439,429 | P314L | *mprF* | Phosphatidylglycerol lysyltransferase | | | ✓ | ✓ | ✓ |
| 2,070,976 | A151T | *vraT* | Regulator of *vraSR* | | | ✓ | ✓ | ✓ |
| 2,233,269 | P114R | *agrB* | Accessory gene regulator B | | | | ✓ | |

maintained $vraT_{A151T}$ but reverted to a wild type *tagH* sequence (T39A). Additionally, BSN14R2 developed a sequence variation in *agrB* (ntC341G, P114R) and a new frameshift mutation in *mprF* (nt745delG, G249Gfs14). BSN14RB maintained $vraT_{A151T}$ but reverted to a wild type *agrB* sequence (R114P) and the intact *mprF* sequence present in BSN14R1 (P314L). Additionally, BSN14RB developed a 5 amino acid in-frame deletion in *saeR* (`nt135 delGATATCATGGTACTT`).

## Patient isolates have a normal baseline mutation rate

One potential rationale for the rapid development of resistance in BSN14 isolates could be an enhanced baseline mutation rate. Elevated mutation frequencies, observed through spontaneous mutations to rifampin *in vitro*, have been shown to contribute to a more rapid resistance development to a number of antibiotics in *S. aureus*, including VAN [11, 12]. The spontaneous rifampin resistance rate of the index isolate from the current encounter (*i.e.* BSN14S1) was determined and compared to the resistance rate of six contemporaneous isolates collected from separate patients as well as to laboratory strain *S. aureus* LAC (Fig 2). Strain BSN14S1 had a median of 27 [interquartile range 23–38] rifampin-resistant mutations per $10^9$ cells compared to 34 [22–63] rifampin-resistant mutations in comparator isolates ($P = 0.12$). These values are consistent with normal baseline mutation frequencies for rifampin resistance in

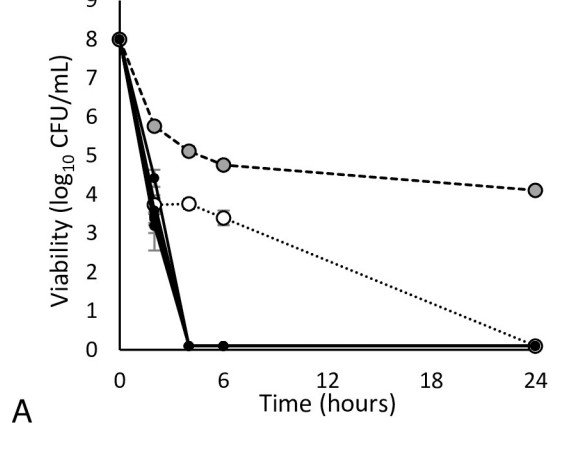

| Patient Isolate | RIF-resistant CFUs | Doubling Time (min) |
|---|---|---|
| C1 | 45 [32-59] | 33 ± 0.7 |
| C2 | 313 [251-358]† | 36 ± 2.1 |
| C3 | 13 [9-20] | 32 ± 2.8 |
| BSN14S1 | 27 [23-38] | 34 ± 0.0 |
| C4 | 28 [26-36] | 36 ± 2.1 |
| C5 | 39 [31-88] | 35 ± 4.9 |
| C6 | 23 [17-51] | 32 ± 2.8 |
| LAC | 40 [32-63] | 29 ± 0.7* |

**Fig 2. Antimicrobial tolerance.** (A) Increased survival of BSN14S1 in daptomycin exposure assay. Cultures in exponential growth phase were adjusted to $10^8$ CFU/mL and exposed to 10 mg/L daptomycin. Data are the means and standard deviations of three independent replicates. Gray markers (dashed lines), BSN14S1; white marker (dotted line), comparator isolate C4. All other replicates were indistinguishable and represented by solid markers and lines. Detection limit, 100 CFU/mL. (B) Number of spontaneous mutations conferring resistance to rifampin per $10^9$ colony-forming units, median [interquartile range]. †Comparator isolate C2 recovered from an unrelated patient is identified as a hypermutator strain ($>100 \times 10^{-9}$). *Strain LAC had a significantly shorter doubling time versus comparators ($P = 0.007$).

staphylococci ($<10^{-7}$) and lower than those seen in strains exhibiting a hypermutable phenotype ($>10^{-7}$) [12–14]. We note that one of the comparator isolates was found to exhibit a hypermutable phenotype and a refinement removing this strain from analysis again demonstrated that the mutation rate in BSN14S1 was unremarkable (31 [21–49] rifampin-resistant mutations in comparator isolates, $P = 0.27$).

### Index patient isolate BSN14S1 is antibiotic tolerant

The rapid development of antibiotic resistance in this patient isolate with an unremarkable mutation rate caused us to suspect a high level of tolerance in the population [2]. DAP exhibits a pronounced difference in its rate of bacterial killing against tolerant staphylococci making it an ideal antibiotic for their identification [15, 16]. Isolate BSN14S1, the six comparators described previously and *S. aureus* LAC were cultivated in liquid culture, exposed to DAP and viability determined at pre-defined intervals. All comparator isolates had a DAP MIC of 0.5 mg/L. Results are provided in Fig 2. Based on standardized definitions, BSN14S1 is antibiotic tolerant at baseline [1]. The fourth comparator isolate (C4) exhibited biphasic killing but only after a 4-log reduction in viability was achieved. Therefore, based on the definitions by Balaban *et al*, none of the comparator strains were tolerant or heterotolerant. While the doubling time of control strain LAC was significantly shorter than that of clinical isolates (29 ± 0.7 m vs. 34 ± 2.5 m, $P = 0.007$), there were no significant differences in doubling time between clinical isolates (Table 2, $P = 0.344$). Therefore, differences in DAP killing were not due to differences in isolate growth rates. Upon further analysis with other anti-MRSA antibiotics, isolate BSN14S1 likewise demonstrated reduced killing by both CPT and VAN at 24 hours compared to MIC-matched comparators (CPT 20 mg/L, 1.0 ± 0.04 vs. 2.2 ± 0.50 log viability reduction, $P < 0.001$; VAN 35 mg/L, 1.7 ± 0.08 vs. 2.2 ± 0.40 log viability reduction, $P = 0.033$). Thus, despite a favorable susceptibility profile based on the organism MIC, antimicrobial tolerance could limit the effectiveness of antistaphylococcal antibiotic therapy and promote the development of antimicrobial resistance.

### Recurrent bacteremia isolates contain mutations in *mprF* and *vraT*

Antimicrobial therapy was initially successful at clearing the MRSA bloodstream infection. However, within three weeks MRSA were again present in surveillance blood cultures. All isolates collected from the patient after initial presentation in October 2018 were genetically

**Table 2. Antimicrobial tolerance.**

| Strain Name | Genetics | DAP | MDK$_{99}$ | TBA | MDK$_{99.99}$ |
|---|---|---|---|---|---|
| C1 | ST3390-MRSA-II CC5. spa t1062. agr type 2 | 0.5 | 1 ± 0.1h | 2 ± 0.1h | 2 ± 0.1h |
| C2 | ST8-MRSA-IVa CC8 spa t008. agr type 1. | 0.5 | 1 ± 0.0h | 1 ± 0.0h | 2 ± 0.0h |
| C3 | ST5-MRSA-IVg CC5 spa t688. agr type 2 | 0.5 | 1 ± 0.0h | 1 ± 0.0h | 2 ± 0.1h |
| BSN14S1 | ST8-MRSA-IVg CC8 spa t064. agr type 1 | 0.5 | 2 ± 0.0h** | 5 ± 0.8h* | 27 ± 2.1h** |
| C4 | ST5-MRSA-II CC5 spa t002. agr type 2 | 0.5 | 1 ± 0.0h | 1 ± 0.1h | 2 ± 0.3h |
| C5 | ST8-MRSA-IVa CC8 spa t008. agr type 1 | 0.5 | 1 ± 0.1h | 1 ± 0.1h | 1 ± 0.2h |
| C6 | ST8-MRSA-IVa CC8. spa t008. agr type 1 | 0.5 | 1 ± 0.0h | 1 ± 0.1h | 2 ± 0.3h |
| LAC | ST8-MRSA-IVa CC8. spa t008. agr type 1 | 0.5 | 1 ± 0.0h | 1 ± 0.1h | 2 ± 0.2h |

Isolates from seven consecutive patients with DAP-susceptible MRSA bacteremia were assessed. BSN14S1 demonstrated significantly prolonged MDK$_{99}$ and MDK$_{99.99}$ values compared to contemporaneously collected patients with identical DAP MICs, indicating tolerance. TBA, time to bactericidal activity, *i.e.* MDK$_{99.9}$.

*$P \leq 0.05$ by one-way ANOVA and post-hoc Student's t-test.

**$P \leq 0.01$ by one-way ANOVA and post-hoc Student's t-test.

related to the initial isolate but now contained mutations in both *mprF* and *vraT*. MprF is a lysylphosphatidylglycerol transferase/flippase that modifies membrane phospholipids with lysine and translocates them to the outer leaflet of the membrane [17]. Mutations in MprF are common in DAP-resistant clinical isolates and result in increased presentation of lysylphosphatidylglycerol on the cell surface and decreased DAP activity [18]. The specific P314L mutation in MprF maps to the flippase domain and was one of the first DAP resistance-conferring mutations identified during the clinical trial resulting in DAP's approval [19]. VraT is a component of the VraTSR three-component regulatory system responsible for regulating cell wall synthesis [20, 21]. Mutations in this system are common in VAN resistant clinical isolates and contribute to both DAP and VAN resistance while conferring collateral susceptibility to β-lactams [22, 23]. The specific A151T mutation identified in VraT is an established contributor to the VISA phenotype and has been identified in reference VISA strains including NRS283 and NRS79 with VAN MICs of 2 and 4, respectively [24]. Consistent with this, most isolates collected during and after the recurrent bacteremia have elevated MICs to both DAP and VAN and population analysis profile (PAP) analysis confirmed that strains with a VAN MIC of 2 had transitioned from VAN-susceptible *S. aureus* (VSSA) to heterogeneous VAN-intermediate resistant *S. aureus* (hVISA).

**The first isolate from the recurrence exhibits impaired TagH activity.** The patient's initial recurrence of bacteremia lasted for four days. In addition to changes to *mprF* and *vraT* discussed above, the first isolate from this recurrence, BSN14R1, had developed a mutation in *tagH*. TagGH/TarGH is a membrane-bound component of the teichoic acid translocation system and the last committed step in wall teichoic acid synthesis. Counterintuitively, decreased TagGH activity can lead to thickened cell walls due to autolysin sequestration [25] but a complete loss of TagGH function is lethal [26]. Thickened cell walls is a common feature of DAP- and VAN-resistant staphylococci [27, 28] resulting from changes to one or more of several global regulators (*graRS*, *vraSR*, *walKR*) or cell wall biosynthetic machinery [29]. In contrast to other potential contributors to cell wall thickening, TagGH activity was specifically associated with the ability of staphylococci to induce biofilm production in the presence of bile components [30]. Therefore, we examined the response of study isolates to biofilm induction with either bovine bile or deoxycholate. Results are presented in Fig 3A. Strain BSN14R1 was unique in its response to challenge, demonstrating no biofilm induction by deoxycholate and a reduction of biofilm in the presence of bile salts. This suggests that the TagH$_{A39T}$ mutation represents a reduction in function. After collection of isolate BSN14R1, the patient's antimicrobial regimen was transitioned from DAP/VAN-based to CPT/LZD-based therapy. This may have selected for the loss of the *tagH* mutation in subsequent isolates as reduced TagGH activity is associated with increased susceptibility to beta lactam antibiotics [31].

**The final isolate from the initial recurrence exhibits impaired MprF activity.** Isolate BSN14R2 was collected on the fourth and final day of the patient's recurrent bacteremia (hospital day 31). In this isolate, a second mutation had occurred in *mprF* resulting in a truncated protein and a reversion from hVISA back to VSSA. Interestingly, the timing of this loss-of-function mutation corresponded to a transition from DAP/VAN-based therapy to CPT/LZD-based therapy. *In vitro* studies suggest that second-site mutations in *mprF* are selected when exposures are switched from DAP-based to β-lactam-based [32, 33]. The identification of an additional frameshift mutation in *mprF* for isolate BSN14R2 was unanticipated as the susceptibility report for this isolate in the patient chart was indistinguishable from the isolate collected three days prior. MprF activity is highly linked to changes in DAP susceptibility and a frameshift mutation would be predicted to decrease DAP resistance [34]. Indeed, repeat susceptibility testing of our BSN14R2 isolate demonstrated markedly different DAP minimum inhibitory concentrations than those reported in the patient record. Isolate BSN14R2 lacking functional

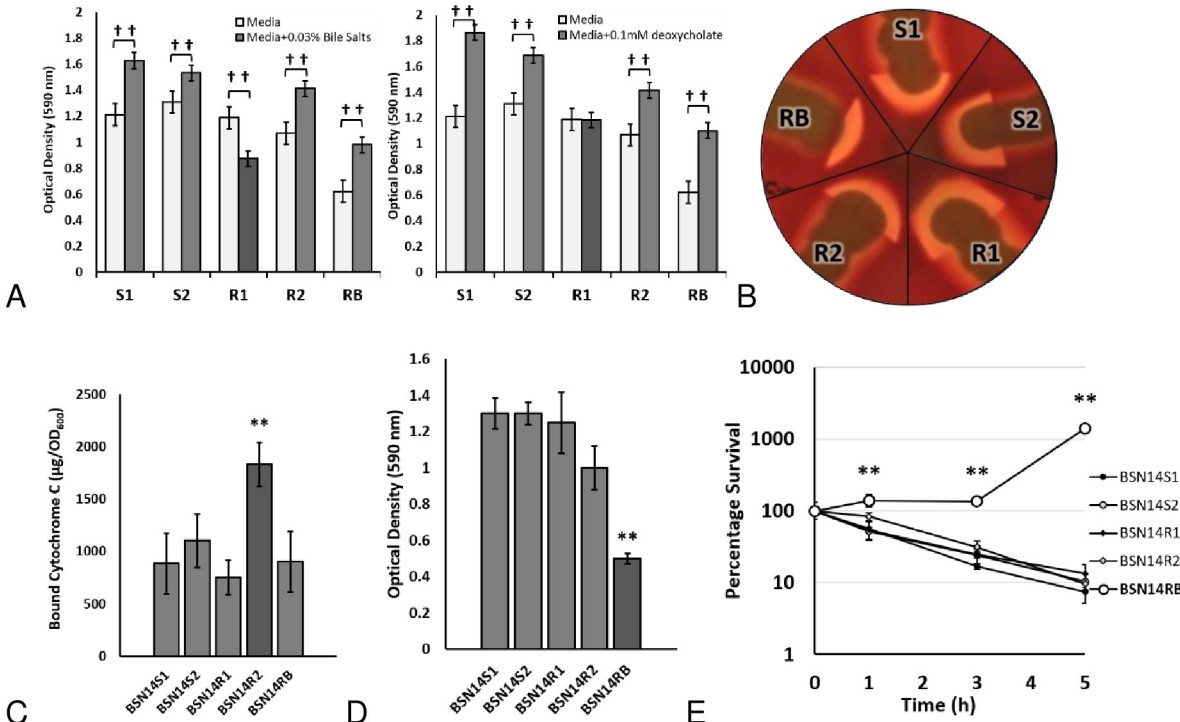

**Fig 3. Phenotypic characterization of BSN14 mutants.** (A) *tagH* mutation present in BSN14R1 alters biofilm production in the presence of bile salts. (B) *agrB* mutation present in BSN14R2 does not alter hemolytic activity. (C) *mprF* mutation present in BSN14R2 alters whole-cell binding of cytochrome C (D) *saeR* mutation present in BSN14RB reduces biofilm production (E) *saeR* mutation present in BSN14RB impairs whole blood killing. $^{\dagger\dagger}$ *P* value < 0.01 compared to same isolate in unsupplemented media. $^{**}$ *P* < 0.01 compared to isolate BSN14S1.

MprF demonstrated a remarkable drop in DAP MIC from 4 mg/L to 0.25 mg/L. MprF is thought to modulate DAP susceptibility by altering the charge on the bacterial cell surface by modification of membrane phospholipids with cationic lysine [18]. BSN14R2 demonstrates significantly more binding of cationic cytochrome C than other isolates suggestive of a more negatively charged cell envelope (Fig 3C). As mentioned previously, this loss of MprF function may again have been selected by the change in pharmacotherapy from a DAP/VAN-based to a CPT-based regimen [32] and the fitness cost of maintaining DAP resistance [35]. In addition to the changes in *mprF*, BSN14R2 contains a mutation in *agrB*. Although AgrB is part of a virulence trait regulon that is frequently mutated in clinical isolates [36–38], the mutation maps to the extracellular interface of a transmembrane alpha helix in a location not thought to be involved in autoinducer binding, processing or transport [39, 40]. Consistent with this, strain BSN14R2 did not exhibit a defect in agr-regulated traits including hemolysis or biofilm production (Fig 3B and 3D) [41].

In order to rationalize the discrepancies between susceptibility values reported in the patient chart and those performed by our group, we repeated CPT, DAP, LZD and VAN MIC testing for all other patient isolates collected. All susceptibility values were within 1 doubling of values reported in the patient chart with the exception of isolate BSN14R2 which consistently demonstrated a low DAP MIC of 0.25 (S2 Table). Our group received a subculture of the isolate used by the clinical lab for susceptibility testing which may not have been representative of the overall population. To assess this, our group identified an investigator that makes similar requests for patient isolates and maintains a separate biorepository of staphylococcal bloodstream isolates. We identified his request for the same isolate, BSN14R2, made on a

separate day from our group. Susceptibility testing of this independent sample collected from the patient on the same day demonstrated the same MICs as performed by our group. Furthermore, we identified duplicate subcultures of BSN14S2, BSN14R1 and BSN14RB. Susceptibility testing and whole genome sequencing of these isolates resulted in indistinguishable MICs and sequences, respectively, to those generated from our collection. We conclude that BSN14R2 represented a mixed population of DAP-susceptible and DAP-resistant bacteria and clinical testing selected for a minority subpopulation that retained the original antibiotic resistant phenotype.

### The first isolate from the second recurrence exhibits altered SaeR regulation

As before, within two weeks of documented negative blood cultures following the initial recurrence, MRSA were again present in the patient's surveillance blood cultures. Isolate BSN14RB, collected hospital day 44 during the patient's second episode of recurrent/relapse bacteremia, contained an internal deletion within the SaeR receiver domain immediately preceding the site of phosphorylation. The SaeR regulon consists of two promoter classes. Class I (high-affinity) promoters regulate factors such as hemolysins and can bind SaeR regardless of phosphorylation status. In contrast, Class II (low-affinity) promoters regulate factors such as coagulase and fibronectin binding protein and require phosphorylated SaeR [42]. As shown in Fig 3B and 3D, BSN14RB has wild-type hemolysin activity but is impaired for biofilm production suggesting its SaeR mutation maintains regulation of Class I promoters but not at Class II. In staphylococci, coagulase production regulated by Class II SaeR promoters decreases survival in human blood [43]. Consistent with this, BSN14RB is uniquely able to survive in heparinated human blood compared to other isolates (Fig 3E). Host factors contribute significantly to the resolution of staphylococcal infection [44]. Staphylococci respond to the presence of neutrophils and defensins by modulating the classical SaeR-regulated production of virulence factors, paradoxically increasing pathogenicity by reducing immune recognition of cytotoxins [45, 46]. Therefore, the five amino acid deletion in SaeR may represent an adaptive trait to promote survival as the organism transitions from an acute to a persistent infection.

### Tolerance accelerates *in vitro* resistance development

The rapid adaptability of BSN14S1 to changing selective pressures supports the body of literature that antibiotic tolerance facilitates the subsequent development of resistance [47]. In order to simulate such selective pressures *in vitro* we subjected BSN14S1 and comparator strains to serial passage in the presence of increasing concentrations of DAP. Each day of serial passage would assess the ability to grow in double the DAP concentration that supported growth on the previous day as DAP resistance typically occurs by the stepwise acquisition of multiple mutations, each contributing to clinically meaningful resistance [48]. Therefore, the minimum time necessary to observe DAP resistance in serial passage (*i.e.* growth in 4 mg/L DAP) would be three days. Replicates of patient isolate BSN14S1 took a median of 3 (range: 3–4) days from growth in 0.5 mg/L DAP to support growth in 4 mg/L DAP. In contrast, comparator isolates took a median of 5 days (range: 4–7) to adapt from growth in 0.5 mg/L DAP to growth in 4 mg/L DAP (Fig 4, $P$ = 0.001). Therefore, despite equivalent basal mutation rates, BSN14S1 develops antibiotic resistance more rapidly than comparator isolates *in vitro*.

## Conclusion

This study links an observed patient case with potential mechanistic understating of how antimicrobial tolerance can facilitate rapid resistance development and adaptation to a new host

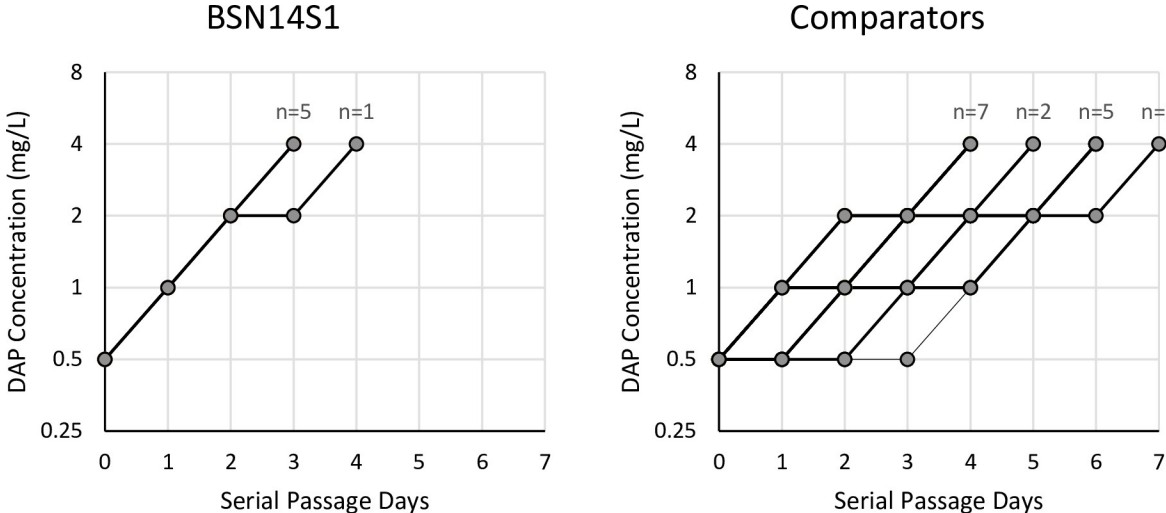

**Fig 4. Daptomycin serial passage.** Time-to-event analysis of tolerant clinical isolate BSN14S1 versus comparator clinical isolates. Comparator strain C2 was excluded from analysis due to its hypermutator phenotype.

environment. Further work is warranted to establish the prevalence and significance of antimicrobial tolerant microbes in resistant and recurrent infection.

## Materials and methods

### Ethics statement

This study was approved by the Detroit Medical Center (IRB 14539) and Wayne State University (IRB 014518M1E). Patient history and clinical course were abstracted from the patient's electronic medical record and patient identifiers removed as outlined in the above IRB approvals. Both review boards provided waivers of informed consent.

### Bacterial isolates, antimicrobials and media

Patient isolates were obtained from the Microbiology Core facility at the Detroit Medical Center. All antibiotics used in this study were purchased commercially as the clinical formulation from West-Ward Pharmaceuticals (Eatontown, NJ, USA, Cefazolin) or Mylan (Canonsburg, PA, USA, Daptomycin and Vancomycin). Activity was confirmed by quality control susceptibility testing against *S. aureus* ATCC 29213 per Clinical and Laboratory Standards Institute (CLSI) guidelines, version M100 ED29:2019 [49]. Mueller-Hinton Broth II (MHB) (BD, Sparks, MD, USA) supplemented with 25 mg/L calcium (as $CaCl_2$) and 12.5 mg/L magnesium (as $MgCl_2$) was used to grow *S. aureus* in liquid culture. All DAP assays used MHB with 50 mg/L calcium as recommended [49]. Population analysis profiling to detect hVISA was performed as described previously using strain Mu3 as the reference standard [50].

### DNA extraction

Isogenic colonies were grown overnight at 37°C to late exponential phase. The cells were pelleted by centrifugation and resuspended in 500 μL SETS buffer (75 mM NaCl, 25 mM EDTA pH 8, 20 mM Tris-HCl pH 7.5, 25% sucrose). RNAse A (10 mg/mL, 5 μL) and lysozyme (25 mg/mL, 10 μL) were added and the sample was incubated at 37°C for 60 min. Proteinase K (20 mg/mL, 14 μL) and 20% SDS (30 μL) were added, the sample was mixed gently by inversion

and incubated at 55˚C for 2 h, inverting occasionally. NaCl (5 M, 200 μL) was added and the sample mixed thoroughly by gentle inversion. Chloroform (500 μL) was then added and the sample mixed by gentle inversion for 30 min at room temperature. Following centrifugation for 15 min at $4,500 \times g$ at room temperature, the upper aqueous phase was transferred to new 1.5 mL tube and another round of chloroform extraction was performed. The upper aqueous phase was transferred to new 1.5 mL tube. The volume was measured and 1/10 that volume of 3 M sodium acetate was added to the sample. DNA was precipitated with 0.7 volumes of iso-propanol and the sample was placed on a slow rocker for 5 min. The filamentous genomic DNA precipitate was fished out with a Pasteur pipette, formed into a hook and sealed with a flame, and transferred to a series of 3 microcentrifuge tubes containing 1 mL 70% ethanol each. The final tube was centrifuged to pellet the DNA and the ethanol was removed with a pipette. The pellet was air dried for several minutes and resuspended in nuclease-free water. A Nanodrop was used to assess the quality of the genomic DNA prep, Qubit BR assay to check the concentration and Agilent TapeStation to check the size distribution.

## Whole genome sequencing

Hybrid assembly of Nanopore MinION and Illumina (150bp PE) reads was performed using Unicycler (v0.4.2) to assemble a complete closed genome. Genomes were annotated using PROKKA (v1.12). Breseq (v0.31.0) was run on BSN14S1 Illumina reads to identify inter-isolate mutations. Default parameters were used for Breseq SNP calling. Sequences have been deposited to GenBank under PRJNA745996.

## Antimicrobial tolerance assays

Study bacteria with identical DAP MICs were adjusted to a McFarland Standard of 0.5 in pre-warmed MHB50 and cultivated with shaking (37˚C, 180rpm) for 1h resulting in an inoculum of $\sim 1 \times 10^8$ CFU/mL. Following the 1 h recovery, DAP was added to a final concentration of 10 mg/L. Samples were removed for colony enumeration via dilution plating immediately prior to addition of DAP and at set intervals after antibiotic challenge. The minimum durations to 2-log, 3-log and 4-log viability reduction ($MDK_{99}$, TBA and $MDK_{99.99}$, respectively) were determined individually per replicate via linear extrapolation between the timepoints immediately preceding and following the indicated $\log_{10}$ unit reduction from baseline. All analyses were performed in triplicate. Between-group differences were assessed by one-way ANOVA and post-hoc Student's t-test. A significant difference in $MDK_{99}$ defines "tolerance" whereas a significant difference in $MDK_{99.99}$ without significantly differing $MDK_{99}$ defines "heterotolerance" [1].

## Mutation rate assays

Seven biological replicates of each strain were cultivated overnight with shaking in 1 mL of Mueller Hinton broth. The number of spontaneously rifampin-resistant colonies were enumerated in triplicate by dilution plating on media containing 25 mg/L rifampin. The number of mutations present per culture were estimated using the Drake formula of the median [51].

## Daptomycin resistance development assay

Three biological replicates of each strain were cultivated overnight with shaking in 1 mL of Mueller Hinton broth. Overnight cultures were subcultured 1:100 into tubes containing 0.25 mg/L or 0.5 mg/L DAP and returned to overnight incubation. Each day the tube with the highest concentration that supported growth was subcultured into tubes containing 1× or 2× DAP

and returned to overnight incubation. Time to DAP resistance was defined as the number of days from growth in 0.5 mg/L DAP until the first day of growth in 4 mg/L DAP. Pairwise comparisons between BSN14S1 and comparators were calculated using the Mann-Whitney U test. An additional independent passage of three biological replicates was performed for BSN14S1 to assess the reproducibility of the findings.

## Phenotypic assays

Bacterial growth rate was determined from serial optical density measurements (600 nm) recorded during exponential growth in Tryptic Soy Broth. Qualitative evaluation of α-, β-, and δ-hemolytic activity was evaluated on Sheep Blood Agar as described previously [41]. *S. aureus* RN4220 was included as a prototypical β-hemolysin-producing strain. Biofilm polystyrene attachment assay was performed in Trypticase Soy Broth with 0.1% dextrose in tissue culture treated 24 well plates (Costar, Corning, NY, USA) and measured using crystal violet as described previously [52]. Biofilm production media was fortified with bovine bile (0.03%, Sigma-Aldrich) or sodium deoxycholate (100μM, Sigma-Aldrich) as indicated [30]. Bacterial survival in heparinated human blood (Zenbio) was determined by dilution plating following 1h and 3h exposure as described previously [43]. Whole-cell binding of cationic cytochrome C was determined spectrophotometrically at 530nm as described previously [53].

## Statistical analysis

DAP MIC results and time-to-event analyses were evaluated using Wilcoxon rank sum test. Two-tailed Student t-test was used for statistical analysis of all other quantitative data. Spearman r was used to determine antibiotic susceptibility correlations. P values of $\leq 0.05$ defined significance.

## Supporting information

**S1 Table. Population analysis profiling.** Vancomycin concentrations, dilutions tested and interpretations were performed based on the method of Sader *et al.* [50]. PAP/AUC ratios (test values relative to Mu3) <0.9, 0.9 to 1.3, and >1.3 are defined as VSSA, hVISA and VISA, respectively. Area under the viability-concentration curve (AUC) was determined using Microsoft Excel software and the trapezoidal method.
(DOCX)

**S2 Table. Confirmatory susceptibility testing.** Minimum inhibitory concentrations were verified for each clinical isolate alongside identical clinical isolates from another biorepository.
(DOCX)

## Acknowledgments

We thank the infectious diseases patient care team at the Detroit Medical Center. We also thank Maureen Taylor at the DMC Core Microbiology lab and Dr. Michael Rybak at the Anti-Infective Research Laboratory for their assistance in obtaining patient isolates.

## Author Contributions

**Conceptualization:** Andrew D. Berti.

**Data curation:** Jonathan M. Monk, Andrew D. Berti.

**Formal analysis:** Jonathan M. Monk, Andrew D. Berti.

**Funding acquisition:** Andrew D. Berti.

**Investigation:** Christopher R. Miller, Jonathan M. Monk.

**Methodology:** Christopher R. Miller, Andrew D. Berti.

**Project administration:** Andrew D. Berti.

**Resources:** Andrew D. Berti.

**Software:** Jonathan M. Monk, Andrew D. Berti.

**Supervision:** Richard Szubin, Andrew D. Berti.

**Validation:** Jonathan M. Monk, Andrew D. Berti.

**Visualization:** Andrew D. Berti.

**Writing – original draft:** Christopher R. Miller.

**Writing – review & editing:** Jonathan M. Monk, Richard Szubin, Andrew D. Berti.

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
