## [Decision Letter · Decision Letter 0]

3 Jun 2021

PONE-D-21-12554

Rapid resistance development to three antistaphylococcal therapies in antibiotic-tolerant Staphylococcus aureus bacteremia

PLOS ONE

Dear Dr. Berti,

Thank you for submitting your manuscript to PLOS ONE. After careful consideration, we feel that it has merit but does not fully meet PLOS ONE’s publication criteria as it currently stands. Therefore, we invite you to submit a revised version of the manuscript that addresses all the points raised by the reviewers during the review process.

We look forward to receiving your revised manuscript.

Kind regards,

Herminia de Lencastre, Ph.D.

Academic Editor

PLOS ONE

Journal Requirements:

We note that you have stated that you will provide repository information for your data at acceptance. Should your manuscript be accepted for publication, we will hold it until you provide the relevant accession numbers or DOIs necessary to access your data. If you wish to make changes to your Data Availability statement, please describe these changes in your cover letter and we will update your Data Availability statement to reflect the information you provide.

Reviewers' comments:

Reviewer's Responses to Questions

**Comments to the Author**

1. Is the manuscript technically sound, and do the data support the conclusions?

Reviewer #1: Yes

Reviewer #2: Partly

2. Has the statistical analysis been performed appropriately and rigorously? 

Reviewer #1: Yes

Reviewer #2: Yes

3. Have the authors made all data underlying the findings in their manuscript fully available?

Reviewer #1: Yes

Reviewer #2: Yes

4. Is the manuscript presented in an intelligible fashion and written in standard English?

Reviewer #1: Yes

Reviewer #2: Yes

5. Review Comments to the Author

Reviewer #1: The manuscript from Miller et al. documents a clinical case report of rapid resistance development to three MRSA therapies in a patient with methicillin-resistant Staphylococcus aureus bacteremia. Genetic, phenotypic and metabolic studies have been done using blood samples recovered over a six-week clinical course. The study technically sounds and this is an area of increasing interest. Authors attributed the rapid emergence of resistance to the daptomycin tolerant phenotype of the index isolate and ruled out the hypermutator phenotype through the determination of spontaneous mutation rate to rifampin in vitro. They were able to justify the appearance of mutations as a result of the antibiotic therapy that was used. It is missing from the manuscript some important information, namely: if the nostrils of the patient have been screened for staphylococcal colonization during the clinical course; the number of colonies picked at each time point and the number of colonies sequenced at each time point.

Instead of having a table in Figure 1, authors may consider to create an infographic timeline combining the information of that table with the antibiotic therapy used and including in the timeline the intermediate blood surveillance cultures negative for MRSA.

Minor points of criticism:

- In the abstract section, line 28 please substitute “the isolates demonstrated” by “the index isolate demonstrated”.

- Please substitute in line 108 “Patients isolates are” by “BSN14S1 is”.

- In results section, line 114 please substitute “All patient isolates” by “All comparator isolates”

- Please state the name to which the acronym BL in line 143 refers to.

- What do the asterisks in Table 2 mean?

- It is missing from materials and methods the conditions of the DNA extraction for WGS.

- Did the authors used default parameters for SNP calling? If yes please state it.

Reviewer #2: This manuscript reports the case of a patient with methicillin-resistant Staphylococcus aureus (MRSA) bacteremia which developed resistance to several antibiotics classically used for MRSA infections (ceftaroline and daptomycin) under the course of treatment (over a period of 44 days). Isolates also showed high minimal inhibitory concentrations (MICs) for vancomycin. Five different isolates were obtained on days 1, 3, 28, 31 and 44. These isolates were sequenced using both Nanopore MinION and Illumina. All isolates were confirmed to be isogenic. Several mutations were present in recurrent isolates compared to the initial one, in genes already described in literature as associated with resistance to daptomycin or heteroresistance to glycopeptides. Then the authors explored several hypotheses to explain the ability of this S. aureus strain to develop these mutations and to understand the potential role of the mutations detected by whole-genome sequencing (WGS). Interestingly they showed that this strain was antibiotic tolerant.

WGS sequences have been deposited to Genbank.

Major comments

- The results of this study are difficult to extrapolate since they only concerned one patient case but they provide insights on the potential link between antibiotic tolerance and development of antibiotic resistance. Nevertheless, data presented here did not allow to affirm that antibiotic tolerance facilitates rapid resistance development.

- In my opinion, the manuscript is not easy to read, probably due to the fact that authors have chosen not to separate results and discussion, and it is difficult to understand what is really new in these results from that is already described in the literature. For instance, the authors described mutations in mprf or in vraT but they did not precise if the exact mutations present in this patient’s isolates have already been described and associated with antibiotic resistance or not.

- Data already published about the link between antibiotic tolerance and development of antibiotic resistance should be more developed by the authors in the introduction and in the discussion. The authors should highlight in the manuscript which of their data provide new insights in the field.

- The authors should provide the complete antimicrobial susceptibility testing profile of the isolates (and not only oxacillin, vancomycin, ceftaroline, daptomycin, linezolid).

- They report the isolates as resistant to vancomycin but isolates have a MIC equal to 2 mg/L, which is susceptible according to CLSI recommendations. If the authors considered the isolates as heteroGISA, they should have performed population analysis profile to confirm this. So the title of the manuscript seems not adapted if the authors included vancomycin in the three antibiotics concerned by development of resistance. Moreover, authors described the appearance of a vraT mutation but the vancomycin MIC of the isolates did not increase over time so it is difficult to assign a role in antibiotic resistance to this mutation.

- In the “Patient case” section, the authors first described the antimicrobial treatments received by the patient and then the results of bacteriological cultures. As the choice of antibiotic therapy depends on the bacteriological results, these latter should be presented before or with antibiotic therapies. Instead of figure 1 presented as a table, a chronological timeline including both bacteriological results and changes in antibiotic therapies would better illustrate the patient case.

- Isolate BSN14RB showed a decreased ceftaroline MIC compared to the previous isolates: did the authors have a hypothesis about this? The authors do not discuss what would be the mechanism of resistance to ceftaroline in isolates R1 and R2.

- The method used to study the binding of cytochrome C is not described in the section “Materials and Methods”.

- The authors described the selection of mutations potentially involved in the development of resistance. Did they test several colonies on the initial S. aureus population or the following isolates to check for the absence of a heterogeneous population?

Minor comments

- Line 70: precise that bacteria were MRSA

- Line 121: results of doubling time are presented in Figure 2 and not Table 2

- Lines 213, 217, 244: please italicize S. aureus

- In the text, define the abbreviations TBA and MDK

6. PLOS authors have the option to publish the peer review history of their article (what does this mean?). If published, this will include your full peer review and any attached files.

Reviewer #1: No

Reviewer #2: No

---

## [Author Response · Author response to Decision Letter 0]

15 Jul 2021

All nucleotide data are now freely available through PRJNA745996. All other Responses to Reviewers are addressed in the uploaded document.

---

## [Decision Letter · Decision Letter 1]

7 Sep 2021

PONE-D-21-12554R1Rapid resistance development to three antistaphylococcal therapies in antibiotic-tolerant Staphylococcus aureus bacteremiaPLOS ONE

Dear Dr. Berti,

Thank you for submitting your manuscript to PLOS ONE. After careful consideration, we feel that it has merit but does not fully meet PLOS ONE’s publication criteria as it currently stands. Therefore, we invite you to submit a revised version of the manuscript that addresses the additional points raised during the review process by the two reviewers. Please submit your revised manuscript by Oct 22 2021 11:59PM. If you will need more time than this to complete your revisions, please reply to this message or contact the journal office at plosone@plos.org. Please include the following items when submitting your revised manuscript:A rebuttal letter that responds to each point raised by the academic editor and reviewer(s). You should upload this letter as a separate file labeled 'Response to Reviewers'.A marked-up copy of your manuscript that highlights changes made to the original version. You should upload this as a separate file labeled 'Revised Manuscript with Track Changes'.An unmarked version of your revised paper without tracked changes. You should upload this as a separate file labeled 'Manuscript'.If applicable, we recommend that you deposit your laboratory protocols in protocols.io to enhance the reproducibility of your results. Protocols.io assigns your protocol its own identifier (DOI) so that it can be cited independently in the future. For instructions see: https://journals.plos.org/plosone/s/submission-guidelines#loc-laboratory-protocols. Additionally, PLOS ONE offers an option for publishing peer-reviewed Lab Protocol articles, which describe protocols hosted on protocols.io. Read more information on sharing protocols at https://plos.org/protocols?utm_medium=editorial-email&utm_source=authorletters&utm_campaign=protocols.

We look forward to receiving your revised manuscript.

Kind regards,

Herminia de Lencastre, Ph.D.

Academic Editor

PLOS ONE

Journal Requirements:

Additional Editor Comments (if provided):

Reviewers' comments:

Reviewer's Responses to Questions

**Comments to the Author**

1. If the authors have adequately addressed your comments raised in a previous round of review and you feel that this manuscript is now acceptable for publication, you may indicate that here to bypass the “Comments to the Author” section, enter your conflict of interest statement in the “Confidential to Editor” section, and submit your "Accept" recommendation.

Reviewer #1: (No Response)

Reviewer #2: All comments have been addressed

2. Is the manuscript technically sound, and do the data support the conclusions?

Reviewer #1: Yes

Reviewer #2: Yes

3. Has the statistical analysis been performed appropriately and rigorously? 

Reviewer #1: Yes

Reviewer #2: Yes

4. Have the authors made all data underlying the findings in their manuscript fully available?

Reviewer #1: No

Reviewer #2: Yes

5. Is the manuscript presented in an intelligible fashion and written in standard English?

Reviewer #1: Yes

Reviewer #2: Yes

6. Review Comments to the Author

Reviewer #1: Authors have addressed reviewers' comments and they still have added additional in vitro experiments to support their arguments, namely, that antimicrobial tolerance may promote the development of antimicrobial resistance. There are still some changes authors can made in this version of the manuscript for its improvement.

1. Lines 78-79 - Please clarify. Based on Figure 1 after the 11th day and for two weeks there was no weekly surveillance cultures.

2. Line 84 - "Blood cultures cleared day 31", shouldn't be day 33?

3. Authors refer that they perform PAP analysis but these results are not shown.

4. Please avoid referring to the isolates by the month of their isolation. Please use instead the day of the isolation or the name of the isolate to be in accordance with Figure 1.

5. Lines 289-291 - Only volumes used were provided, no reference for the concentration of the solutions used is provided.

6. Line 332 - "0.25 mg/L or 0.5 mg/L" of what?

7. Figure 1 - Please define antibiotics' abbreviations in the legend.

8. Figure 4 - In material and methods is referred that three biological replicates of each strain have been used. As so, why are represented 6 replicates (n=6) for BSN14S1?

Reviewer #2: The authors have addressed all the points raised during the first review. Thus the clarity and robustness of the manuscript’s conclusions have been strongly improved.

I have some minor comments.

- Line 19: index isolate is not susceptible to trimethoprim/sulfamethoxazole so change “all (non-beta-lactam) antibiotics” for “almost all” or “all… except trimethoprim/sulfamethoxazole”

- Line 23: isolate BSN14RB is not susceptible to ceftaroline (table S1, MIC=2, as previous isolates R1 and R2)

- Line 93: genetic variations are not indicated in Figure 1

- Line 111: please indicate that data are presented In Figure 2. Results in the text are not consistent with those in Figure 2 (23-38 vs 23-28 for BSN14S1)

- Line 216: data are shown in Table S2

- Line 332: precise that tubes contained daptomycin

- Figure 1: what does ⴕ mean in the table?

- Table S1: if the table is published, it would be useful to add the date of collection of each isolate as in Figure 1.

7. PLOS authors have the option to publish the peer review history of their article (what does this mean?). If published, this will include your full peer review and any attached files.

Reviewer #1: No

Reviewer #2: No

---

## [Author Response · Author response to Decision Letter 1]

7 Sep 2021

Reviewer #1: Authors have addressed reviewers' comments and they still have added additional in vitro experiments to support their arguments, namely, that antimicrobial tolerance may promote the development of antimicrobial resistance. There are still some changes authors can made in this version of the manuscript for its improvement.

We thank the Reviewer for the additional recommendations and hope that we have adequately addressed them in this Revision.

1. Lines 78-79 - Please clarify. Based on Figure 1 after the 11th day and for two weeks there was no weekly surveillance cultures.

You are correct, weekly surveillance cultures were not performed as recommended in the patient notes. We have revised line 79 replacing “on weekly surveillance cultures” to read as “on subsequent cultures (days 10 and 11)”

2. Line 84 - "Blood cultures cleared day 31", shouldn't be day 33?

You are correct, we had mistakenly indicated day 31, the last day of culture positivity, as the first day of cleared blood culture. We have revised this line to indicate the first day of documented culture clearance as day 33.

3. Authors refer that they perform PAP analysis but these results are not shown.

We have added Supplemental Table 1 to explicitly provide the PAP analysis results. 

4. Please avoid referring to the isolates by the month of their isolation. Please use instead the day of the isolation or the name of the isolate to be in accordance with Figure 1.

 We have revised throughout to avoid use of months when referencing isolates.

5. Lines 289-291 - Only volumes used were provided, no reference for the concentration of the solutions used is provided.

 Concentrations of reagents have been added

6. Line 332 - "0.25 mg/L or 0.5 mg/L" of what?

These were concentrations of daptomycin and the word had been removed in error. We thank the reviewer for catching this omission and the abbreviation “DAP” has been added as intended.

7. Figure 1 - Please define antibiotics' abbreviations in the legend.

 Antibiotic abbreviations have been defined as requested.

8. Figure 4 - In material and methods is referred that three biological replicates of each strain have been used. As so, why are represented 6 replicates (n=6) for BSN14S1?

In order to increase the rigor of the comparisons, after isolate BSN14S1 demonstrated a more rapid emergence of DAP resistance than comparators, we performed a second, independent analysis in triplicate from a separate freezer stock of the strain. This replicated experiment was consistent with the earlier findings of more rapid emergence of DAP resistance. This demonstrates that the initial findings are reproducible.

Reviewer #2: The authors have addressed all the points raised during the first review. Thus the clarity and robustness of the manuscript’s conclusions have been strongly improved.

We thank the Reviewer for the kind words and hope these additional changes further strengthen the manuscript.

I have some minor comments.

- Line 19: index isolate is not susceptible to trimethoprim/sulfamethoxazole so change “all (non-beta-lactam) antibiotics” for “almost all” or “all… except trimethoprim/sulfamethoxazole”

We agree and have amended the abstract to include SXT resistance in the strain description.

- Line 23: isolate BSN14RB is not susceptible to ceftaroline (table S1, MIC=2, as previous isolates R1 and R2)

You are absolutely correct and we apologize for the oversight. Our first E-test for Ceftaroline in this isolate indicated an MIC of 1 for this isolate and we based our initial abstract on these data. Upon repeat testing in triplicate, the MIC for BSN14RB was 2, 2, 1.5 but we neglected to update the abstract. We thank the Reviewer and have corrected this error.

- Line 93: genetic variations are not indicated in Figure 1

We agree that revised Figure 1 no longer contains genetic data and have updated the text to only refer to Table 1.

- Line 111: please indicate that data are presented In Figure 2. Results in the text are not consistent with those in Figure 2 (23-38 vs 23-28 for BSN14S1)

Thank you for catching that typographical error. The correct numbers are in the text (23-38). Figure 2 has been updated to provide the correct distribution (23-38).

- Line 216: data are shown in Table S2

The “data not shown” has been updated to point toward supplemental table S2.

- Line 332: precise that tubes contained daptomycin

The text has been corrected.

- Figure 1: what does ⴕ mean in the table?

The superscript data has been defined as confirmation by E-test

- Table S1: if the table is published, it would be useful to add the date of collection of each isolate as in Figure 1.

Hospital day of collection has been added to the new Table S2 as requested.

---

## [Decision Letter · Decision Letter 2]

23 Sep 2021

PONE-D-21-12554R2Rapid resistance development to three antistaphylococcal therapies in antibiotic-tolerant Staphylococcus aureus bacteremiaPLOS ONE

Dear Dr. Berti,

Thank you for submitting your manuscript to PLOS ONE. After careful consideration, we feel that it has merit but does not fully meet PLOS ONE’s publication criteria as it currently stands. Therefore, we invite you to submit a revised version of the manuscript that addresses the additional points raised by reviewer #1.

We look forward to receiving your revised manuscript.

Kind regards,

Herminia de Lencastre, Ph.D.

Academic Editor

PLOS ONE

Journal Requirements:

Additional Editor Comments (if provided):

Reviewers' comments:

Reviewer's Responses to Questions

**Comments to the Author**

1. If the authors have adequately addressed your comments raised in a previous round of review and you feel that this manuscript is now acceptable for publication, you may indicate that here to bypass the “Comments to the Author” section, enter your conflict of interest statement in the “Confidential to Editor” section, and submit your "Accept" recommendation.

Reviewer #1: All comments have been addressed

2. Is the manuscript technically sound, and do the data support the conclusions?

Reviewer #1: Yes

3. Has the statistical analysis been performed appropriately and rigorously? 

Reviewer #1: Yes

4. Have the authors made all data underlying the findings in their manuscript fully available?

Reviewer #1: Yes

5. Is the manuscript presented in an intelligible fashion and written in standard English?

Reviewer #1: Yes

6. Review Comments to the Author

Reviewer #1: The authors have addressed all the comments raised during the previous round of review. However, I still have some minor comments:

- There is a typo in the beginning of the title of the manuscript. Please remove the letter "z".

- Please clarify the susceptibility profile of the blood cultures mentioned in lines 22-23. In accordance to figure 1 and table S1, the blood culture remained with intermediate resistance to vancomycin and resistant to daptomycin and ceftaroline susceptibility was not available for this blood culture in the patient chart.

- Line 160 - Please remove the word "in".

- The interpretation criteria used to define vancomycin phenotype in table S1 (VSSA, hVISA or VISA) should be provided, as well as the PAP graphics. In table S1 why was isolate BSN14S2 considered as VSSA and isolates BSN14R1 and BSN14RB as hVISA?

- Please provide the name of the software used for the determination of AUC.

7. PLOS authors have the option to publish the peer review history of their article (what does this mean?). If published, this will include your full peer review and any attached files.

Reviewer #1: No

---

## [Author Response · Author response to Decision Letter 2]

23 Sep 2021

Reviewer #1: The authors have addressed all the comments raised during the previous round of review. However, I still have some minor comments:

1. There is a typo in the beginning of the title of the manuscript. Please remove the letter "z".

This typo has been corrected.

2. Please clarify the susceptibility profile of the blood cultures mentioned in lines 22-23. In accordance to figure 1 and table S1, the blood culture remained with intermediate resistance to vancomycin and resistant to daptomycin and ceftaroline susceptibility was not available for this blood culture in the patient chart.

We agree with the reviewer that it is difficult to convey susceptibility profiles to multiple antibiotics and multiple isolates in a concise narrative format. We agree that the information available to clinicians indicated probable VAN heteroresistance, DAP resistance and an unknown CPT susceptibility profile for the “RB” isolate. Ceftaroline susceptibility by Etest was ordered but the patient had been discharged before completion of the add-on testing and this test was canceled. Subsequent testing performed in table S1 confirmed the CPT intermediate resistance indicated in the text. While the ceftaroline susceptibility profile was not known to clinicians at the time, ceftaroline intermediate resistance was highly suspected due to previous cultures and confirmed by our group in table S1. Lines 23-24 have been modified to provide more precise descriptions.

- Line 160 - Please remove the word "in".

 This typo has been corrected.

- The interpretation criteria used to define vancomycin phenotype in table S1 (VSSA, hVISA or VISA) should be provided, as well as the PAP graphics. In table S1 why was isolate BSN14S2 considered as VSSA and isolates BSN14R1 and BSN14RB as hVISA?

Interpretive criteria for PAP testing is a standard defined by the reference cited in the methods and in the table legend. We have reproduced the definition in the table legend. PAP graphics are not typically provided in manuscript analogous to photographs of Etest strips not typically provided in manuscripts. We have added a PAP graphic for review but do not feel it is informative for publication. Excel software was used to determine trapezoidal AUC. We note that the order of the data presented in table S1 previously was in error and thank the Reviewer for catching this oversight. The AUCs were not accurately presented to the reader. This has been corrected in the revised manuscript.

- Please provide the name of the software used for the determination of AUC.

We used Excel to calculate the AUC based on the formula (yn+1+yn)/2 × (xn+1-xn) where “y” is the log-transformed CFU/ml measured at consecutive concentrations of vancomycin “x”. This series of trapezoidal AUCs were then summated across the measured concentrations.

---

## [Decision Letter · Decision Letter 3]

1 Oct 2021

Rapid resistance development to three antistaphylococcal therapies in antibiotic-tolerant Staphylococcus aureus bacteremia

PONE-D-21-12554R3

Dear Dr. Berti,

We’re pleased to inform you that your manuscript has been judged scientifically suitable for publication and will be formally accepted for publication once it meets all outstanding technical requirements.

Kind regards,

Herminia de Lencastre, Ph.D.

Academic Editor

PLOS ONE

Additional Editor Comments (optional):

Reviewers' comments:

Reviewer's Responses to Questions

**Comments to the Author**

1. If the authors have adequately addressed your comments raised in a previous round of review and you feel that this manuscript is now acceptable for publication, you may indicate that here to bypass the “Comments to the Author” section, enter your conflict of interest statement in the “Confidential to Editor” section, and submit your "Accept" recommendation.

Reviewer #1: All comments have been addressed

2. Is the manuscript technically sound, and do the data support the conclusions?

Reviewer #1: (No Response)

3. Has the statistical analysis been performed appropriately and rigorously? 

Reviewer #1: (No Response)

4. Have the authors made all data underlying the findings in their manuscript fully available?

Reviewer #1: (No Response)

5. Is the manuscript presented in an intelligible fashion and written in standard English?

Reviewer #1: (No Response)

6. Review Comments to the Author

Reviewer #1: Authors have changed the order of the AUCs data presented in table S1, however they did not change the values of "PAP/AUC ratio" in the second line of the table accordingly. Please revise that line too.

7. PLOS authors have the option to publish the peer review history of their article (what does this mean?). If published, this will include your full peer review and any attached files.

Reviewer #1: No

---

## [Editor Report · Acceptance letter]

11 Oct 2021

PONE-D-21-12554R3 

Rapid Resistance Development to Three Antistaphylococcal Therapies in Antibiotic-Tolerant *Staphylococcus aureus* Bacteremia 

Dear Dr. Berti:

I'm pleased to inform you that your manuscript has been deemed suitable for publication in PLOS ONE. Congratulations! Your manuscript is now with our production department. 

Kind regards, 

on behalf of

Dr. Herminia de Lencastre 

Academic Editor

PLOS ONE